# Lessons learned from operationalizing the integration of nutrition-specific and nutrition-sensitive interventions in rural Ethiopia

Girmay Ayana Mersha[1,2]*, Eshetu Zerihun Tariku[2,3], Wanzahun Godana Boynito[2,3], Meseret Woldeyohaness[1], Tadese Kebebe[1], Birhanu Wodajo[1], Stefaan De Henauw[2], Souheila Abbeddou[2]

1 Food Science and Nutrition Research Department, Ethiopian Public Health Institute, Addis Ababa, Ethiopia, 2 Public Health Nutrition, Department of Public Health and Primary Care, Ghent University, Ghent, Belgium, 3 School of Public Health, Arba Minch University, Arba Minch, Ethiopia

* GirmayAyana.Mersha@UGent.be

## Abstract

### Background

Undernutrition reduction requires coordinated efforts across sectors to address its causes. A multisectoral approach is important in diagnosing the problem and identifying solutions that would be implemented across different sectors.

The study aimed to explore the experience of health and agriculture extension workers in integrating nutrition-specific and nutrition-sensitive interventions provided to households with children under two years of age at community level.

### Methods

A qualitative study was conducted in agrarian areas of Ethiopia following the completion of a multi-sectoral program that integrated health and agriculture interventions, in 2021. The program's goal was to reduce stunting and improve the dietary diversity of young children. In total, 28 key informant interviews were conducted with health- and agriculture-extension workers and mothers. A framework analysis approach was applied to manage and analyze data using NVivo version 12 software.

### Results

The study showed that joint health and agriculture interventions improved community knowledge on childcare and agricultural practices. The practice of farm gardening and cooking demonstrations were improved after the implementation of the program. Because of service integration, extension workers perceived an improved father's role in supporting mothers in childcaring and feeding nutritious diets to children and decreased severe cases of undernutrition. Integration of health and agriculture sectors for nutrition intervention was challenged by the high workload on extension workers, poor supervision and leadership

**Data availability statement:** The data is available at the following qualitative data repository link: DOI 10.5281/zenodo.14884554.

**Funding:** We received funds from Global Minds Fund of Ghent University https://www.ugent.be/en/research/funding/devcoop/globalmindsfund.htm (GRANT GMF.CAB.2020.0025.01) and the Ethiopian public health institute. The funders had no role in study design, data collection and analysis, decision to publish or preparation of the manuscript.

**Competing interests:** The authors have declared that no competing interests exist.

commitment, lack of appropriate agricultural inputs, and absence of clarity on sector-specific roles. In some areas nutrition services are not owned by the health and agriculture sectors, and it was overlooked.

## Conclusion

Integrating community-level platforms were key entry points to address undernutrition and promote key agriculture and health interventions. The joint implementation of health and agriculture services were effective in the reduction of wasting and improved the role of family members in supporting mothers. However, Integration of sectors were facing challenges in creating shared vision to improve nutrition status of children and women, distributing workload and equal commitment among sectors.

## Introduction

Undernutrition including stunting, wasting, underweight and micronutrient deficiencies plagued populations in low- and middle-income countries for decades and remains one of the most pressing health problems [1]. Due to their increased nutrient requirements young children are vulnerable to undernutrition. Additionally, women of reproductive age are vulnerable to undernutrition because of their lower social status, which limits their access to nutrient-rich foods [2]. The causes of undernutrition are complex and affected by factors acting at different causal hierarchical levels [3]. The political, economy, health, education, agriculture and food systems, water and sanitation, and the environment are all determinants of maternal and child nutrition and health [4]. Undernutrition reduction requires coordinated efforts across sectors to address its underlying causes [5, 6]. Addressing the problem of undernutrition needs the collaboration of multiple sectors and a variety of stakeholders in governments, non-government organizations, and the private sector [7]. Undernutrition reduction requires a departure from a health system-centered approach of provision of nutritional services and nutrition-specific interventions only, to an integrated approach that encompasses nutrition-sensitive, food systems-based approaches[8].

The framework of actions proposed by the UNICEF and the Lancet nutrition series recognizes the needs for multiple actions that are at different levels from immediate, underlying to distal [1,3]. Interventions targeting the improvement of maternal and child nutrition and health act on inter-linked factors which influence each other. Nutrition-specific interventions address the immediate causes of undernutrition and nutrition-sensitive interventions focus on the underlying causes of good nutrition [9]. Both nutrition-sensitive and specific interventions require an enabling environment, that encompasses actions addressing basic determinants of undernutrition. Multiple actions to improve nutrition were undertaken over the recent years, the most prominent is Scaling Up Nutrition which clearly underscores the necessity of actions among all the stakeholders [10]. This approach puts nutrition at the core of each sector plan and institutional effort.

However, systematic coordination between different sectors to achieve the objectives of improved maternal and child nutrition is problematic, given the bureaucratic barriers that characterize the administrative division of responsibilities among sectors[6]. All sectors are expected to be clear about what they can contribute to resolving malnutrition in a particular context specific to their sector role [11]. Agriculture, as an example of nutrition-sensitive intervention, can potentially increase food availability, which would directly improve the dietary accessibly leading to improved nutritional status of targeted populations [12]. However, the multiple factors that influence nutrition, the social dynamics within households that

can result in unexpected allocations of resources, and the many indirect ways that agriculture may impact nutrition, require purposeful planning and careful consideration of how exactly program inputs will lead to nutritional improvements [13]. This complex relationships has consequently lead to limited impact of nutrition intervention, because other determinants are not put in place at different levels [14].

Ethiopia is one of the countries with the highest rate of undernutrition in sub-Saharan Africa. The proportions of stunting, underweight and wasting among children under age of five years are respectively 38%, 24% and 10% in Ethiopia [15]. The reduction of undernutrition remained stagnant, with the prevalence of stunting (37%), underweight (21%), and wasting(7%), in 2019 [16]. Multiple efforts have been made to implement the government commitment towards ending undernutrition. To reach this goal, Ethiopia has demonstrated a strong policy commitment to nutrition through the development of a multisectoral food and nutrition policy, which highlights the need to accelerate multi-sectoral approaches [17]. Political willingness is an important factor that helps to implement effective nutrition interventions [10]. However, the Ethiopian situation remains in its debuts with poorly synchronized sectors of food, nutrition, agriculture, and health to prevent the problem of under-nutrition. In Ethiopia, the presence of health and agriculture extension systems and frontline workers at the community level is a good framework that facilitates the synergetic linkage of nutrition within health and agriculture services [18] Additionally, existing multisectoral nutrition coordination structure at all levels, and the willingness of international and national development partners to support multisectoral nutrition interventions are favorable conditions on the ground [19]. Ethiopia has implemented multiple nutrition-specific and nutrition-sensitive programs such as the Ethiopian national nutrition program, and the Seqota declaration program, that have required the development of structures and processes that facilitate multisectoral coordination [20]. But, collaborations between the health and agriculture sectors are suboptimal at community levels [21]. The multi-sectoral nutrition programs and interventions lack strong coordination that brings on board all the relevant stakeholders.

Based on these facts, Ethiopia designed different intervention modalities whose objective is to strengthen community-level multisectoral nutrition interventions. In this context, the Sustainable Undernutrition Reduction program in Ethiopia (SURE) was developed and implemented between 2016 and 2020 in rural regions of Ethiopia. The program utilizes the potentials of the Agricultural Development Agents (DAs) and the Health Extension Workers (HEWs) for the promotion of improved agricultural practices and utilization of high nutrient products to promote dietary diversification among rural mothers and children.

A finding from a coverage survey of the SURE program indicates the interventions had a positive effect on improving minimum diet diversity, minimum acceptable diet and complementary feeding practices in children [22]. At present, there is no study that has analyzed the experience and perception of the health (HEWs) and agriculture extension workers (AEWs) in integrating nutrition services provision at the community level in rural Ethiopia. Understanding frontline workers' practices and experience is crucial for identifying appropriate measures to sustain nutrition improvements [9]. This qualitative study aims at appraising the understanding, the knowledge and perception of those women and men who are the most in contact with the communities in rural Ethiopia and who were responsible to implement the SURE program.

## Methods

### Study setting

The Ethiopian Ministry of Health and the Ministry of Agriculture led the implementation of the SURE program whose aim was to reduce stunting by improving complementary feeding

practices and dietary diversity of young children through a process of improving local agricultural productivity, food accessibility, food consumption diversity and nutritional status of young children and women [27]. SURE, was implemented between 2016 and 2020 in four agrarian regions (Oromia, Amhara, Southern Nations, Nationalities, and Peoples' Region [SNNPR], and Tigray).

The program package comprised both nutrition-specific and nutrition-sensitive interventions. The nutrition-specific interventions included child feeding practices, dietary diversity, and the treatment of childhood illnesses. On the other hand, the nutrition-sensitive interventions focused on promoting diverse food production, water, sanitation, and hygiene (WASH), and women's empowerment, all of which support improved nutrition outcomes. The SURE intervention districts were selected because of the high food insecurity and stunting prevalence in four regions out of the nine total regions in Ethiopia. Details of the SURE intervention program have been published elsewhere [23].

Briefly, a total of 36 districts (Woreda as locally called) were selected and assigned to the SURE intervention. Eligible households are households with at least one child under the age of 2 years. The intervention started in 2016, 2146 households from 144 Kebeles (smallest administrative unit) were included.. After baseline data collection, and all households received counselling at household and community levels. Additionally, 10% of households with children aged under two years were 1) provided improved seed variety and chicken through AEWs and 2) trained through demonstration gardens at farmer training centers.

The SURE program was delivered through: 1) interpersonal contacts to provide counselling on IYCF and nutrition-sensitive agriculture advice to mothers and fathers of children under 24 months, inclusive of pregnant women and fathers-to-be, jointly delivered by the local HEWs and AEWs during routine household visits; 2) men's and women's group dialogues targeting all men and all women in a given community network, facilitated also by the local HEWs and AEWs, and 3) media campaign to reinforce IYCF and dietary diversity messages.

## Study design

A phenomenological qualitative study design was conducted among beneficiaries (mothers) and implementers (HEWs and AEWs) of the SURE program in the intervention communities. One Kebele per district from the intervention group was randomly selected to contribute in the qualitative study.

## Study period

This qualitative study was conducted after four years of intervention. The data were collected at the community level in selected districts of Amhara, Oromia, and SNNPR between January and February 2021. Data collection was discontinued in Tigray because of the war.

## Recruitment of participants

All SURE intervention districts and Kebeles in the three agrarian regions of Ethiopia were eligible for inclusion. The recruitment of participants was done purposely to obtain maximum variation of information. The HEWs and AEWs as service providers of the intervention packages, and mothers who were enrolled in the SURE intervention as direct beneficiaries were invited to participate in this study. The HEWs and AEWs were eligible to participate in this study if they received the SURE program training in 2016 and have provided the service as per their training in their respective Kebele for a minimum of two years. Mothers should be residents of the Kebele, would have had a child 0–24 months old at the time the intervention started, and received joint health and agriculture counseling as described in the study setting.

## Data collection and tools

Data were collected using key informant interviews (KIIs) at the community level from a total of 28 eligible respondents including HEWs, AEWs and mothers. Participants were asked about their understanding of nutrition, quality of service provided and quality of nutrition education specific to the SURE program, the challenges of implementing nutrition programs jointly, and how the program could be sustained for a better nutrition service provision (Supplementary material 1).

Participants took part in a face-to-face interview. A topic guide was developed by the research team and training on how to conduct KIIs was provided. Supervisors were assigned to each region to coordinate the interviews and ensure that all procedures were followed to obtain accurate and relevant information. The interview began with demographic questions about schooling, experience, and professional background. All the interviews were conducted in a private space and were audio recorded. The adequacy of the sample size was ensured by obtaining maximum information from both implementers' and beneficiaries' perspectives until saturation was achieved. Data was collected in the local languages (Amharic and Oromifa). Audio recordings were transcribed and translated into English. All transcribed data was anonymized.

## Data analysis

A framework analysis approach was used to manage and analyze the data. Framework approach provides clear steps to follow and produces structured outputs of summarized data that can be used with a variety of data sources, including interviews, focus groups, and documents [24]. The steps include familiarization (becoming familiar with the content of the data), Identification of thematic areas (identifying a thematic framework, which is a set of predetermined themes or categories that are relevant to the research question), Indexing (systematically coding the data to identify content related to each theme or category), and Interpretation (identifying patterns and relationships across the data, and interpreting what these patterns and relationships mean in relation to the research question). Qualitative data analysis software NVIVO version 12 software was used in coding and constructing thematic areas from the collected data. The transcripts were coded into specific themes. The themes were identified and coded as causes of undernutrition in the community, challenges of integrating nutrition services and lessons learned in implementation of nutrition-specific and nutrition-sensitive programs.

## Ethical considerations

Ethical approval was obtained from the research ethic committees both at the University Hospital of Ghent, Belgium (protocol number BC-08862), and the Ethiopian Public Health Institute, Ethiopia (protocol number EPHI-IRB-306-2020). The qualitative study included recipients from the same cohort (participating mothers from the intervention group), in addition to health extension workers from the intervention Kebeles. Written informed consents, which included permission to be recorded using a digital audio recorder, were obtained from participants before interviews. Further administrative permissions were granted by local government bureaus in all regions.

# Results

## Demographic characteristics of study participants

From the total of twenty-eight key informants who participated in the study, 43%, 39% and 18% were from Oromia, Amhara and SNNPR regions, respectively. The proportion of HEWs,

AEWs and mothers were almost similar (Table 1). Totally 58% of the extension workers were female and 73% had at least five years of work experience. Sixty-seven percent of the study participants have completed college and above.

The main thematic areas of the analysis included the causes of undernutrition in the community, the challenges of integrating health and agriculture services, and lessons learned from implementing the joint program. Each thematic area was further divided into subthemes.

For the causes of undernutrition, subthemes included factors such as land and water shortages, drought, lack of irrigation, diseases, and poor hygiene and sanitation. Under the challenges, subthemes addressed issues like civil unrest and war, the COVID-19 pandemic, shortages of agricultural inputs, workload, program ownership, training, and technical support. In the lessons learned, subthemes focused on knowledge gained, monitoring and evaluation practices, and the effectiveness of farm and cooking demonstrations.

## Major causes of undernutrition

The key informants perceived that, in their community, undernutrition was due to a shortage of land and water to produce adequate food for a rapidly growing population.

*Farmland is inadequate, with farmers owning an average of 0.5 hectares and it is difficult to produce enough food for family consumption using one farming land. Moreover, rain shortage and lack of irrigation service, caused undernutrition in our community. AEW,SNNPR and HEW, Oromia.*

**Table 1. Demographic characteristics of the qualitative study participants, Ethiopia.**

| Demographic Characteristic(N = 28) | Frequency (%) |
|---|---|
| **Age of participants (in years)** | |
| 20–30 | 16 (57.1) |
| 31–40 | 10 (35.7) |
| 41–50 | 2 (7.1) |
| **Region** | |
| Amhara | 11 (39.3) |
| Oromia | 12 (42.9) |
| SNNPR | 5 (17.9) |
| **Role** | |
| Agriculture extension worker | 9 (32.1) |
| Health extension worker | 10 (35.1) |
| Mother | 9 (32.1) |
| **Sex of participants** | |
| Female | 20 (71.4) |
| Male | 8 (28.6) |
| **Health and agriculture extension workers experience** | |
| 2–4 years | 5 (26.3) |
| 5–10 years | 7 (36.8) |
| >10 years | 7 (36.8) |
| **Education status of participants** | |
| No formal education | 3 (10.7) |
| Primary school | 6 (21.4) |
| Technical/vocational college | 15 (53.6) |
| Diploma and above | 4 (14.3) |

Frequent drought and absence of irrigation facilities to farm during the dry season are among the causes of undernutrition that undermine the efforts of AEWs and households to improve food and nutrition interventions.

*Malnutrition in this area is primarily caused by insufficient fruit and vegetable production due to drought. As a result, our community has struggled to produce fruit and vegetables and has given them less priority. HEW, Oromia.*

An extension worker indicated that the causes of undernutrition are related to lack of hygiene and sanitation, diseases and infections, inflation and war that displaced people from their residential areas and hindered them from producing food.

*The causes of undernutrition include poor hygiene, disease, war, and food security issues. In some areas, market inflation has worsened the situation. Many families can no longer afford essential items like milk, leading to malnutrition in children. HEW, SNNPR.*

*Women who rely on market-based livelihoods often introduce other foods to their infants before six months and leave them at home, as they cannot afford to stay home and breastfeed regularly. HEW, Oromia.*

The COVID pandemic have also compromised the supportive supervision and monitoring activities by restricting people's movement from regional and zonal levels to front line workers.

*Our team used to meet frequently, with discussions documented in meeting minutes. Initially, these meetings were held weekly, then shifted to monthly. We have stopped gathering participants for training sessions because such activities conflict with COVID-19 restrictions. HEW, Oromia.*

### Challenges of integrating health and agriculture interventions in Ethiopia

Many challenges have been reported by extension workers in the implementation of intersectoral interventions. High workload, conflict, shortage of agricultural inputs, limited training and poor supportive supervision and COVID-19 pandemic were the most frequently reported challenges (Fig 1).

### Intervention in the midst of the war in Ethiopia

Ethiopia is facing civil unrest in the last four years and the project intervention has been affected. Conflict was a challenge of service delivery at community level, which resulted in destruction of facilities, damages of means, and loss of income by restricting people's movement and targeted attacks on workers. Achieving political stability has become increasingly challenging.

*The current situation is highly sensitive. People are becoming increasingly anxious, feeling that the situation is escalating into something worse.*

*At the start of the SURE program, an expert was present to support the community daily. People were receiving education and training, which encouraged them to adopt and practice new skills. However, now, many have abandoned these efforts and turned their focus to politics. – AEW, SNNPR*

### Shortage of agricultural inputs and material for workshop training

The availability of agricultural inputs for gardening and cooking demonstrations at community level was a challenge and participants were not interested in bringing their own food and material for demonstration. The project gives agricultural inputs to only 10% of the total population, but the demand for agriculture inputs was high and the community perceived that the project does not treat people equally.

*We are very interested in conducting food preparation demonstrations, but a lack of access to necessary ingredients poses a challenge. Asking farmers to provide the ingredients themselves leads to a loss of interest in participating. AEW, Oromia.*

### Workload and ownership of the program

Nutrition activities are not prioritized equally across sectors, with unclear ownership and extension experts viewing them as extra tasks. AEWs lack awareness and commitment to

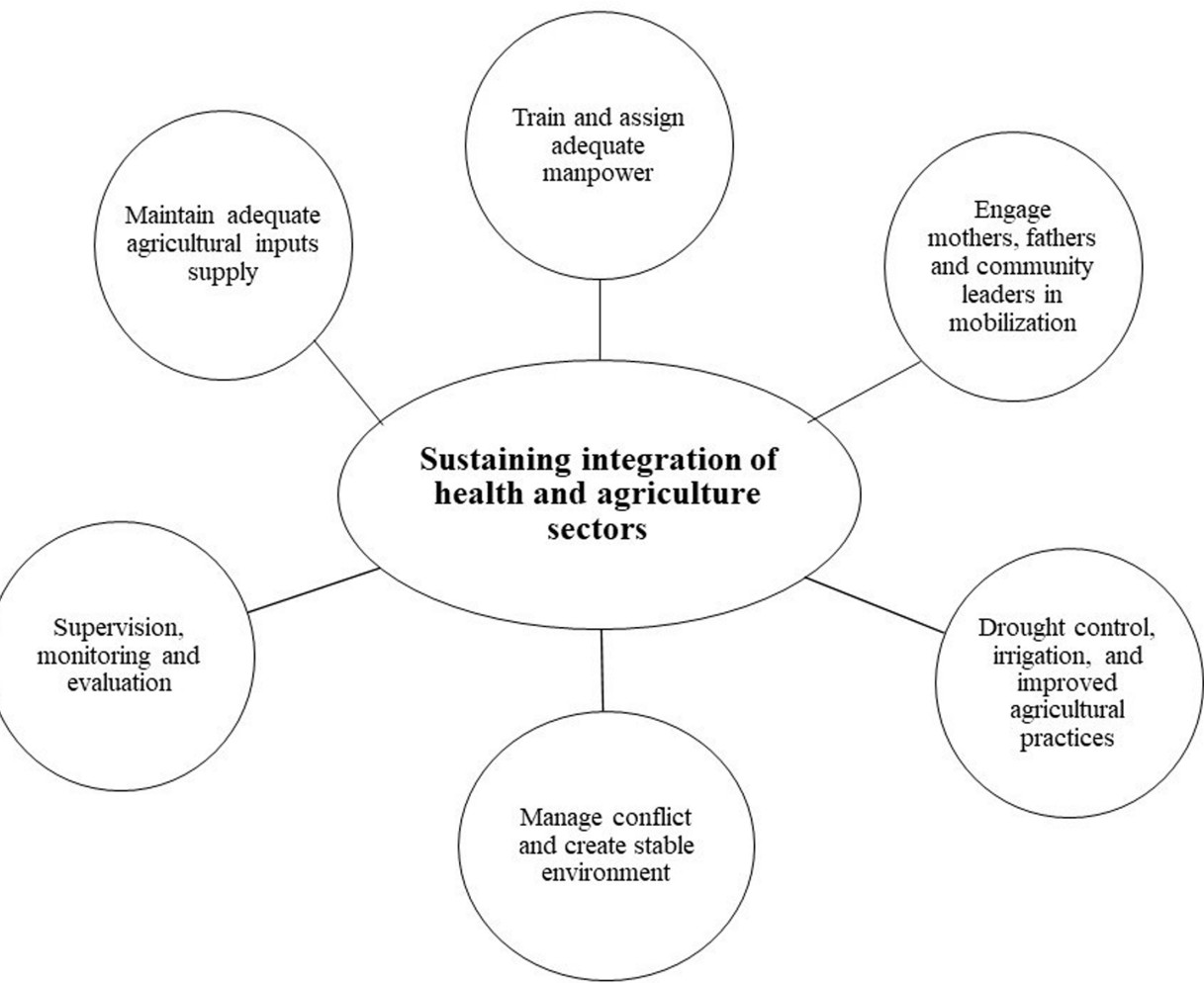

**Fig 1. Sustaining the integration of agriculture and health sectors for reduction of undernutrition in Ethiopia.**

implementing nutrition at the community level, assuming it is the responsibility of HEWs to address women's and children's nutrition.

> *There are often seasonal jobs, campaigns on vaccination, planting, harvesting, and irrigation. Appointments are missed as priority is given to main tasks, especially in health, where immunization and follow-up programs have increased. AEW, SNNPR.*

Integrating agriculture and health services faces workforce shortages and a lack of prioritization for food and nutrition interventions. Leadership focuses on sector-specific campaigns, such as vaccinations in health and water conservation in agriculture. Extension workers see nutrition activities as requiring dedicated experts, and the service is further hindered by the lack of continuous monitoring and evaluation mechanisms.

> *The main challenge lies in being overburdened and dealing with overlapping responsibilities. Agricultural extension workers often assume SURE program activities are not part of their duties and conversely, the health extension argues that these responsibilities fall under agriculture domain. As a result, we focus solely on our primary responsibilities and expect others to handle the SURE program tasks. HEW, Oromia.*

### Absence of adequate training, supervision, and technical support

Supportive supervision and technical support during the implementation of new program is a key factor for the success of a project. The SURE program was designed in a way that supervisors contribute continuously during the whole implementation period. But, the support provided was limited and many extension workers perceived the support and supervision provided were not adequate.

> *If the relevant authorities regularly visited our activities and provided feedback, it would motivate us to improve further. Active involvement from the zonal level would be highly beneficial. but this seems to be an overlooked activity. HEW, Oromia.*

> *When support is well-organized and delivered hierarchically to the kebele level, farmers are receptive and eagerly adapt it to their households. If the support appears poorly coordinated, the community dismisses it as useless. AEW, Oromia.*

### Lesson learned and sustainability of health and agriculture intersectoral intervention

The SURE intervention packages depended on multi sectoral coordination at the local and district levels (Fig 1). However most agricultural interventions were perceived as the distribution of poultry and other agricultural materials. Although most extension workers captured the intention of the target population, as being mothers, young children and those who are most vulnerable households.

> *The SURE program focused on distributing up to six hens per household, targeting low-income parents, lactating mothers, and children under two. It also provided vegetable seeds and training for professionals and communities. AEW, Amhara.*

> *After 15 days of initial training, the program was implemented at the community level, distributing chickens to 20 individuals per kebele and supplying agricultural tools like water pumps to support backyard fruit and vegetable production. AEW, Oromia.*

Health and agriculture extension workers jointly visited households with under two years of age children and provided age-appropriate IYCF counselling and advice on nutrition-sensitive agriculture to improve dietary diversity. Job aids have been developed with food calendars and recipe books for each region. Extension workers describe their experience of joint visit as follows:

> In the SURE program, we collaborate with the kebele's men's and women's development teams to support health and agriculture extension efforts. Together with health extension workers, we visit households, teaching families, especially pregnant mothers and children under two, how to grow and use vegetables and livestock for home consumption. We guide them on production, planting, and feeding practices, emphasizing diverse and nutritious diets. AEW, SNNPR.

Mothers witnessed the service received by extension workers in their community. Although, asking mothers to come with food ingredients could be one factor that limits mothers from attending cooking demonstration sessions.

> From the HEW, I learned about preparing porridge using food items readily available at home. I also learned the importance of exclusively feeding children breast milk until the age of six months.

> From the AEW, I gained knowledge on the importance of having vegetables in our gardens and animal products at home. If these items are not available, we are encouraged to buy them from the market and prepare porridge using a variety of these nutritious food items to feed our children. – A mother, Oromia

Farm demonstration gardens were established at the farmer's training centers to facilitate coordination between the HEW managing the health posts and the AEWs working on the farms. Farm demonstration gardens were developed and managed by AEWs. Appropriate varieties of vegetables were selected according to agroecology. Harvests from the farmer's training centers were used for cooking demonstrations. Cooking demonstrations were managed by HEWs at health posts near the farmer's training centers. Women also donate ingredients from their homes to contribute to the exercise.

> We give agricultural demonstrations on every occasion. Whenever we get the best performance, we give training and demonstrations on farmer's land or on farmer's training centers to act as the best performed farmers. We also give demonstrations as per the farmers' needs and questions. AEW, Amhara.

> When they advised us, they said in recent days, even not the rural, the urban communities are practicing home gardening with the limited land they have in the backyard. Different types of vegetables (tomatoes, kale, potatoes...) are on your hand; you may not need to buy them from the market, plant them. Mother, Oromia.

## Knowledge gained through the SURE program

The SURE program interventions have impact on the knowledge of community to improve child caring practice, agricultural and behavioral practices in different parameters like severe undernutrition workload, women empowerment, and father's support. The extension workers have observed the changes in the community, mainly fathers' role in supporting mothers in

childcaring and the importance given to feeding nutritious diet to children instead of selling the food produced. The extension workers felt that severe cases of undernourished children visiting heath centers have decreased.

*… After implementing the SURE program, we noticed a significant decline in the prevalence of malnutrition. The number of children presenting at our health post with malnutrition has significantly declined. Prior to the program, malnutrition cases were more common, and many required referrals. Cases of severe malnutrition have become exceedingly rare since the program's introduction.*

*Regarding women's empowerment, one of the most significant changes brought by the SURE program is the active involvement of husbands in food demonstrations. This has encouraged them to recognize their shared responsibility in childcare and support for their wives. A notable outcome has been the initiation of community-wide discussions on various women's issues. With continued counseling and support, significant improvements in these areas have been observed. – HEW, Oromia*

*In the past, men believed that childcare was solely a woman's responsibility. However, this perspective has completely changed; men now actively support their wives by helping care for their children. – AEW, SNNPR.*

### Importance of monitoring and regular supervision

Awareness creation activities should be organized frequently for proper implementation of nutrition interventions, and ideally by another person responsible for coordinating nutrition activities at community level. Additionally, the importance of regular meetings and evaluation for integrating services at community level was highlighted.

*The first step is providing awareness-creation education, a responsibility currently assigned to HEWs and AEWs. However, as these workers are often occupied with other tasks, it is essential to designate another responsible body to handle this role. This would help reduce workload and prevent task overlapping among HEWs and AEWs, thereby improving the effectiveness of the SURE program.Another important aspect is conducting regular meetings to monitor and evaluate the program's progress. All responsible stakeholders should participate in these meetings. AEW, Amhara.*

### Discussion

This qualitative study has explored the causes of undernutrition and the experience of integrating health and agriculture sectors for improved food security and nutritional status in Ethiopia. Most of the participants reported that the major cause of undernutrition in the were drought, conflict, and population pressure. Furthermore, the high burden of infectious diseases and WASH problems were the cause of the undernutrition. Our respondents witnessed health and agriculture sectors joint interventions The practices of farm gardening and cooking demonstrations were improved because of the joint intervention. The intervention has impact on knowledge gain at community level to improve child caring practices, diversified agricultural practices and women empowerment. Father's role in supporting mothers in childcaring and feeding nutritious diets to children has improved, and the case load of wasted children has been decreased. Despite of this successful experience integration of health and agriculture sectors for nutrition intervention was challenged by various factors including lack of training,

poor supervision and leadership commitment, lack of appropriate agricultural inputs, and absence of clarity on sector-specific roles. In some areas nutrition services were not owned by the health and agriculture sectors, and it was overlooked. This qualitative study finding showed nutrition interventions have been perceived as the work which mainly involves health sector only. Furthermore, the study showed extension workers were overloaded by different sectoral activities which can limit their joint efforts and compromise the integration of nutrition-related activities.

The finding in this study is aligned with the study of Mohamed (2017) which found drought, population pressure, and conflict as the major causes of food security problems in Ethiopia, resulting in child undernutrition and household food insecurity [25]. Similarly finding on child growth in the time of drought by Hoddinott and Kinsey (2001) has shown that conflict and drought increase stunting prevalence among young children in poor households [26].

Nutrition interventions require intensive effort from multiple sectors. The expansion of governmental services from nutrition-specific to nutrition-sensitive strategies is the preferred approach, as it addresses all determinants of maternal and child nutrition and health including social and economic determinants [20]. A research conducted in Rwanda among pregnant women and lactating mothers by Michael Habtu et al. (2023) demonstrated that integrating nutrition-specific and nutrition-sensitive intervention had an impact on the improvement of nutrition knowledge and skills, particularly in relation to balanced diets [27]. Although multi-sectoral interventions that involve agriculture, health and other relevant sectors are increasingly recommended, operating at village, district and regional levels within low-income countries is complex [28].

Ethiopia has a national structure for multi sectoral nutrition coordination anchored at the Ministry of Health and a working model at district level also exists [19]. Nevertheless, according to this finding nutrition has been perceived as the work of the Ministry of Health and has been framed as a treatment/preventive problem, which mainly involves health sector interventions. SURE is one of the government-led programs that try to bring the health and agriculture sectors to work jointly for improved dietary diversity and improved nutritional status [23,29]. It is a common umbrella that brings HEWs, AEWs and women to promote dietary diversity and better nutrition in Ethiopia. But we found extension workers are overloaded by different sectoral activities which can limit their joint efforts and compromise the integration of nutrition-related activities. Similar to this study, frontline workers in Uganda confront various pressures from the institutional and socio-political environment, that limit their decisions about their daily activities [9]. Similar to our findings, Pelletier and collaborators found that in sub Saharan African countries, institutional integration of nutrition in different sectors was challenged by the job description of staff, the lack of sectoral alignment on multisectoral nutrition objectives and integration into the planning and reporting of sector activities [19]. A systematic review exploring the integration of nutrition-sensitive and specific interventions in fragile environments shows a lack of interest in collaboration, workload and diverting to only sectoral activities as the barriers to integrating nutrition services [30]. Similarly a study from Burkina Faso has shown the extent of integrating nutrition agenda in to government policy varied from one sector to another[31] which affects successful implementation of multisectoral nutrition interventions.

Delivering nutrition services to the local level tends to work better in countries that have adequate decentralized structures. In Ethiopia, the vertical structure of the Ministry of Health and Ministry of Agriculture from central to Kebele level offers an opportunity to integrate nutrition services at community level. Front-line HEWs and AEWs are a critical bridge between communities and service provision systems in the health and agriculture sectors.

Empowering extension workers, supporting them and responding to the challenges they face will be an important part of ensuring the sustainability of joint activities to promote proper nutrition.

### Strengths and limitations

Our study gathered data from implementers and beneficiaries of the joint intervention program, offering insights for scaling up and refining similar projects. It serves as a model for integrating multiple sectors in future project designs at grassroot level. However, our study lacks information from the subject matter specialists at the district and regional level tasked with providing technical support and supervision of the extension workers. Furthermore, the study did not explore the experience of family members, specifically father's role in caring and child feeding.

However, as our study was conducted at the end of the project, the implementers might provide biased information regarding its sustainability and potential for scaling up. Additionally, the extension workers may perceive it as an added burden to their workload, which could also lead to biased information. Hierarchical level information is needed. Furthermore, our study did not explore the experience of regional and district level coordination offices and family members, specifically father's role in caring and child feeding.

### Conclusion

The study has demonstrated integrating community-level platforms are key entry points to scale up nutrition knowledge and promote key agriculture and health interventions in Ethiopia. Community-based joint interventions are perceived as appropriate approaches to prevent severe cases of undernutrition and improve the role of fathers in supporting mothers. Despite these benefits, joint interventions face challenges in establishing interaction between sectors to create a shared vision in improve nutrition status of children and women, distributing workload and equal commitment among sectors. Empowering the key implementersi.e extension workers through training, monitoring, and supervision, should be in place to ensure the successful implementation of joint programs in the health and agriculture sectors and addressing the challenges they face are part of ensuring the sustainability of integration between sectors.

### Supporting information

**S1 File. Interview guide.**
(DOCX)

**S2 File. Checklists for reporting qualitative studies.**
(DOCX)

AcknowledgmentWe are grateful to the study participants. In addition, our appreciation goes to the health officials from central to community level, health, and agriculture extension workers for facilitating the implementation of the study. Finally, we would like to thank our data collectors and supervisors for their support.

### Author contributions

**Conceptualization:** Girmay Ayana Mersha, Stefaan De Henauw, Souheila Abbeddou.

**Data curation:** Girmay Ayana Mersha, Eshetu Zerihun Tariku, Meseret Woldeyohaness.

**Formal analysis:** Girmay Ayana Mersha, Souheila Abbeddou.

**Funding acquisition:** Stefaan De Henauw, Souheila Abbeddou.

**Investigation:** Girmay Ayana Mersha, Eshetu Zerihun Tariku, Meseret Woldeyohaness, Tadese Kebebe, Birhanu Wodajo.

**Methodology:** Girmay Ayana Mersha, Wanzahun Godana Boynito, Tadese Kebebe, Stefaan De Henauw, Souheila Abbeddou.

**Project administration:** Girmay Ayana Mersha, Eshetu Zerihun Tariku, Meseret Woldeyohaness, Birhanu Wodajo, Stefaan De Henauw, Souheila Abbeddou.

**Resources:** Stefaan De Henauw, Souheila Abbeddou.

**Software:** Girmay Ayana Mersha.

**Supervision:** Girmay Ayana Mersha, Eshetu Zerihun Tariku, Meseret Woldeyohaness, Tadese Kebebe, Birhanu Wodajo, Souheila Abbeddou.

**Validation:** Girmay Ayana Mersha, Souheila Abbeddou.

**Visualization:** Girmay Ayana Mersha, Souheila Abbeddou.

**Writing – original draft:** Girmay Ayana Mersha.

**Writing – review & editing:** Eshetu Zerihun Tariku, Wanzahun Godana Boynito, Meseret Woldeyohaness, Tadese Kebebe, Birhanu Wodajo, Stefaan De Henauw, Souheila Abbeddou.

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
