## [Decision Letter · Decision Letter 0]

20 Aug 2024

PONE-D-23-25004Lessons learned from operationalizing the integration of nutrition-specific and nutrition-sensitive interventions in rural EthiopiaPLOS ONE

Dear Dr. Mersha,

Thank you for submitting your manuscript to PLOS ONE. After careful consideration, we feel that it has merit but does not fully meet PLOS ONE’s publication criteria as it currently stands. Therefore, we invite you to submit a revised version of the manuscript that addresses the points raised during the review process.

**ACADEMIC EDITOR: In addition to review comments, address the following: **

There are grammer and language flaws that needs intense revisions. Go throughout the entire document and make revisions. Briefly explain how you ensured adequacy of the sample size?How you ensured the quality of qualitative data in this study?Briefly describe who were the data collectors and supervisors? 

We look forward to receiving your revised manuscript.

Kind regards,

Dawit Wolde Daka

Academic Editor

PLOS ONE

Reviewers' comments:

Reviewer's Responses to Questions

**Comments to the Author**

1. Is the manuscript technically sound, and do the data support the conclusions?

Reviewer #1: Partly

2. Has the statistical analysis been performed appropriately and rigorously? 

Reviewer #1: Yes

3. Have the authors made all data underlying the findings in their manuscript fully available?

Reviewer #1: Yes

4. Is the manuscript presented in an intelligible fashion and written in standard English?

Reviewer #1: Yes

5. Review Comments to the Author

Reviewer #1: Abstract:

• Use proper tenses e.g instead of the study aims, it should be the study aimed; A qualitative study has been conducted should read ….. study was conducted

• Define briefly what was the large intervention program

• The first two sentences of the result are describing the intervention rather than the findings of the study

• The authors claims that the study was done in 4 agrarian areas but one region was removed due to war. In the description of the intervention can be 4 but this study was done in 3 agrarian areas

Background

• Provide citations for statement from line 43 to 45

• The following statement should be revised into two sentences ‘Young children and women of reproductive age are particularly vulnerable because of their increased nutrient requirements, and, in case of women, their lower social status, which limits their access to nutrient-rich foods’.

• The third paragraph (lanes 68 -80), the authors claim some facts but no citations in the whole paragraph. These claims should be supported with citations

• On line 83, EDHS 2016 was used for prevalence of stunting, underweight and wasting while there is EDHS 2019. Do the authors want to refer before the intervention?

• I was just curious as to if there had been any quantitative evaluations of the Sustainable Undernutrition Reduction Program in Ethiopia (SURE) program published; if so, it would be preferable to include the resluts in the background.

Methods

• Authors should clearly describe what was/were the component(s) for nutrition-specific intervention and for nutrition-sensitive intervention used under the Sustainable Undernutrition Reduction program in Ethiopia (SURE). Could the authors more clearly identify for the reader which intervention parts are nutrition-specific versus those that are nutrition-sensitive?

• It is not clear why authors are including control districts in the method section instead of description of the intervention districts

• What was the justification of selecting one Kebele per district randomly? Do you believe the findings would have been different if you had chosen the Kebeles with the highest or lowest percentage of maternal or child malnutrition?

• The intervention started in 2016 and data collection was conducted in 2021, so the data collection was done after 5 years not 4 years.

• How were the 28 Key Informants sampled/reached? According to Table 1 there are 9 Agriculture extension workers, 10 Health extension workers and 9 Mothers. For example, what were the basis or criteria to select 9 mothers purposively in each Kebele? Explanation is needed also how out of 36 districts arrived to those numbers?

Results

• Overall comment: the findings are not aligned to the topic of the study. There some themes, not related to the lessons learned from the intervention. E.g Causes of malnutrition …. Does it mean that they came to know the causes after intervention???

• What were the perceived positive impact of the intervention? What were the challenges faced during intervention? Are there any recommendations proposed by the participants?

• So, it is recommended that, first authors should come up with a Table indicating the themes and sub-themes. This will facilitate accurate analysis and interpretation while preventing muddle and confusion as it is now.

Discussion

• Is paragraph 1, relevant to the topic and objective of the study?

• The discussion is not discussing point by point according to the findings. Authors should re-write this section by aligning to the key themes and sub-themes.

• The authors should discuss the literature of other similar interventions (such as SUAAHARA, ASTUTE, https://doi.org/10.1016/j.cdnut.2022.100018, etc.) and fit their findings into the body of knowledge already available on implementation research for these significant multi-sectoral programs and what this study adds to that body of knowledge.

Strengths and limitations

• This section is poorly written and seems the inclusion and exclusion criteria.

• Authors could be more thoughtful with the limitations, both with the participants and the bias from the program implementers

Conclusion

• Not well written to reflect the findings. What recommendations can be given to address the challenges to enhance impact and uptake of future programs?

Overall comments

• Adjust the typos and grammatical errors throughout the document

6. PLOS authors have the option to publish the peer review history of their article (what does this mean?). If published, this will include your full peer review and any attached files.

Reviewer #1: **Yes: **Dr. Michael Habtu

---

## [Author Response · Author response to Decision Letter 1]

3 Oct 2024

TO: The Editor, Plos one

Dear Sir/ Madam:

We would like to take this opportunity to thank you for considering our manuscript for publication in you journal and we appreciate your comments and suggestions which certainly improved our manuscript.

As requested, we have addressed all comments made by the editor and reviewer, revised the manuscript and tracked the changes we made, and provided you with point-by-point response for the comments made. The response to reviewers is attached separately o the submersion portal.

We believe, and as acknowledged by the reviewers, our manuscript contributes to the existing knowledge gap by sharing lessons learned from a joint program aimed to provide nutritional services to the community. Therefore, we kindly request you to progress our manuscript towards publication in your journal.

---

## [Decision Letter · Decision Letter 1]

30 Oct 2024

PONE-D-23-25004R1Lessons learned from operationalizing the integration of nutrition-specific and nutrition-sensitive interventions in rural EthiopiaPLOS ONE

Dear Dr. Mersha,

Thank you for submitting your manuscript to PLOS ONE. After careful consideration of reviewer comments, we invite you to satisfactorily address and submit a revised version of the manuscript that addresses the points raised during the review process.

We look forward to receiving your revised manuscript.

Kind regards,

Dawit Wolde Daka, PhD

Academic Editor

PLOS ONE

Reviewers' comments:

Reviewer's Responses to Questions

**Comments to the Author**

1. If the authors have adequately addressed your comments raised in a previous round of review and you feel that this manuscript is now acceptable for publication, you may indicate that here to bypass the “Comments to the Author” section, enter your conflict of interest statement in the “Confidential to Editor” section, and submit your "Accept" recommendation.

Reviewer #1: (No Response)

2. Is the manuscript technically sound, and do the data support the conclusions?

Reviewer #1: Partly

3. Has the statistical analysis been performed appropriately and rigorously? 

Reviewer #1: N/A

4. Have the authors made all data underlying the findings in their manuscript fully available?

Reviewer #1: Yes

5. Is the manuscript presented in an intelligible fashion and written in standard English?

Reviewer #1: No

6. Review Comments to the Author

Reviewer #1: The authors made an effort to respond to the comments, but some were not fully addressed.

Abstract

• To be more specific and reflecting the topic: the integration of nutrition-specific and nutrition-sensitive interventions should incorporated in the background of the abstract. For instance, “The study aimed to explore the experience of health and agriculture extension workers in integrating nutrition-specific and nutrition-sensitive interventions provided to households…….

Methods

• Again, to reflect the topic authors should clearly outline what was/were the component(s) for nutrition-specific intervention and for nutrition-sensitive intervention used under the Sustainable Undernutrition Reduction program in Ethiopia (SURE). They were indicated in the response of point-by-point but not in the manuscript.

Results

• The result section was not adequately addressed as per the comments given before. For example, authors responded, “We have learned that support of family members in childcare and low incidence of severe cases of undernutrition were among the positive effects of the project in the community. The presence of high workload on extension workers, civil unrest, shortage of agricultural inputs, limited training and poor supportive supervision and COVID-19 pandemic were the most frequently reported challenges mentioned by the study participants. They have suggested to strengthen training, assign appropriate man-power, adequate budget for implementation and close supervisions were recommendations forwarded by the participants.” however, these are not reflected in the result.

• Aside from the theme of challenges, the results are not clearly presented. After outlining the socio-demographic characteristics, the authors should explicitly highlight the main themes and sub-themes to enhance reader comprehension.

• The theme of "Lessons learned and sustainability of the health and agriculture intersectoral intervention" is mentioned, but there are no clear key lessons provided that could be applied to other settings.

• The data analysis procedure was well described but not aligned with result.

• Some of the quotations are quite lengthy, making them difficult to follow. It would be helpful to either shorten them or include additional, more concise quotations for clarity.

Discussion

• The discussion is not discussing point by point according to the findings. Authors should re-write this section by aligning to the key themes.

7. PLOS authors have the option to publish the peer review history of their article (what does this mean?). If published, this will include your full peer review and any attached files.

Reviewer #1: **Yes: **Michael Habtu

---

## [Author Response · Author response to Decision Letter 2]

9 Dec 2024

We have addressed the reviewer’s comments point by point, as detailed in the attached documents.

---

## [Decision Letter · Decision Letter 2]

3 Jan 2025

Lessons learned from operationalizing the integration of nutrition-specific and nutrition-sensitive interventions in rural Ethiopia

PONE-D-23-25004R2

Dear Dr. Mersha,

We’re pleased to inform you that your manuscript has been judged scientifically suitable for publication and will be formally accepted for publication once it meets all outstanding technical requirements.

Kind regards,

Dr.Dawit Wolde Daka, PhD

Academic Editor

PLOS ONE

Additional Editor Comments (optional):

Reviewers' comments:

Reviewer's Responses to Questions

**Comments to the Author**

1. If the authors have adequately addressed your comments raised in a previous round of review and you feel that this manuscript is now acceptable for publication, you may indicate that here to bypass the “Comments to the Author” section, enter your conflict of interest statement in the “Confidential to Editor” section, and submit your "Accept" recommendation.

Reviewer #1: All comments have been addressed

2. Is the manuscript technically sound, and do the data support the conclusions?

Reviewer #1: Yes

3. Has the statistical analysis been performed appropriately and rigorously? 

Reviewer #1: Yes

4. Have the authors made all data underlying the findings in their manuscript fully available?

Reviewer #1: (No Response)

5. Is the manuscript presented in an intelligible fashion and written in standard English?

Reviewer #1: Yes

6. Review Comments to the Author

Reviewer #1: (No Response)

7. PLOS authors have the option to publish the peer review history of their article (what does this mean?). If published, this will include your full peer review and any attached files.

Reviewer #1: **Yes: **Michael Habtu

---

## [Editor Report · Acceptance letter]

PONE-D-23-25004R2

PLOS ONE

Dear Dr. Mersha,

I'm pleased to inform you that your manuscript has been deemed suitable for publication in PLOS ONE. Congratulations! Your manuscript is now being handed over to our production team.

Kind regards,

on behalf of

Mr Dawit Wolde Daka

Academic Editor

PLOS ONE